# Suppressive Effects of Volatile Compounds from *Bacillus* spp. on *Magnaporthe oryzae Triticum* (MoT) Pathotype, Causal Agent of Wheat Blast

**DOI:** 10.3390/microorganisms11051291

**Published:** 2023-05-16

**Authors:** Musrat Zahan Surovy, Shahinoor Rahman, Michael Rostás, Tofazzal Islam, Andreas von Tiedemann

**Affiliations:** 1Division of Plant Pathology and Crop Protection, Department of Crop Sciences, Georg-August-University of Goettingen, Grisebachstrasse 6, 37077 Goettingen, Germany; 2Institute of Biotechnology and Genetic Engineering (IBGE), Bangabandhu Sheikh Mujibur Rahman Agricultural University, Salna, Gazipur 1706, Bangladesh; tofazzalislam@bsmrau.edu.bd; 3Division of Agricultural Entomology, Department of Crop Sciences, Georg-August-University of Goettingen, Grisebachstrasse 6, 37077 Goettingen, Germany; shahinoor_ent@yahoo.com (S.R.); michael.rostas@uni-goettingen.de (M.R.)

**Keywords:** *Bacillus*, biocontrol, volatile organic compound, GC–MS, sporulation

## Abstract

The *Magnaporthe oryzae Triticum* (MoT) pathotype is the causal agent of wheat blast, which has caused significant economic losses and threatens wheat production in South America, Asia, and Africa. Three bacterial strains from rice and wheat seeds (*B. subtilis* BTS-3, *B. velezensis* BTS-4, and *B. velezensis* BTLK6A) were used to explore the antifungal effects of volatile organic compounds (VOCs) of *Bacillus* spp. as a potential biocontrol mechanism against MoT. All bacterial treatments significantly inhibited both the mycelial growth and sporulation of MoT in vitro. We found that this inhibition was caused by *Bacillus* VOCs in a dose-dependent manner. In addition, biocontrol assays using detached wheat leaves infected with MoT showed reduced leaf lesions and sporulation compared to the untreated control. VOCs from *B. velezensis* BTS-4 alone or a consortium (mixture of *B. subtilis* BTS-3, *B. velezensis* BTS-4, and *B. velezensis* BTLK6A) of treatments consistently suppressed MoT in vitro and in vivo. Compared to the untreated control, VOCs from BTS-4 and the *Bacillus* consortium reduced MoT lesions in vivo by 85% and 81.25%, respectively. A total of thirty-nine VOCs (from nine different VOC groups) from four *Bacillus* treatments were identified by gas chromatography–mass spectrometry (GC–MS), of which 11 were produced in all *Bacillus* treatments. Alcohols, fatty acids, ketones, aldehydes, and S-containing compounds were detected in all four bacterial treatments. In vitro assays using pure VOCs revealed that hexanoic acid, 2-methylbutanoic acid, and phenylethyl alcohol are potential VOCs emitted by *Bacillus* spp. that are suppressive for MoT. The minimum inhibitory concentrations for MoT sporulation were 250 mM for phenylethyl alcohol and 500 mM for 2-methylbutanoic acid and hexanoic acid. Therefore, our results indicate that VOCs from *Bacillus* spp. are effective compounds to suppress the growth and sporulation of MoT. Understanding the MoT sporulation reduction mechanisms exerted by *Bacillus* VOCs may provide novel options to manage the further spread of wheat blast by spores.

## 1. Introduction

Wheat is a major cereal crop worldwide [1], and, according to the United States Department of Agriculture (USDA), 779.03 million tons of wheat were produced globally in 2021 [2]. Wheat blast is a devastating fungal disease that is caused by the *Magnaporthe oryzae Triticum* (MoT) pathotype [3,4,5]. It can cause significant reductions in wheat yield and grain quality [6,7]. Wheat blast first emerged in Brazil in 1985 and then gradually spread to Argentina, Bolivia, and Paraguay [5,6,8]. Outbreaks of wheat blast in Bangladesh and Zambia in recent years confirmed its expansive potential [9,10]. Recently, it has also been recognized as a potential future threat to European wheat production [11].

Fungicides are commonly used to control blast and other ear-related wheat diseases [12]. However, the failure of fungicides to fully control MoT has led to repetitive applications, increasing the selection pressure on the pathogen to develop resistance [13,14]. The emergence of new fungicide-resistant strains of MoT severely threatens wheat production [7,15], which is why the use of resistant cultivars is an important strategy to control wheat blast. However, MoT can break cultivar resistance, and, so far, no cultivars are entirely resistant to MoT [5,16].

Biological control is a potential alternative to manage plant pathogens [17,18,19]. Over the past three decades, multidisciplinary research in biocontrol has investigated effective strategies to control wheat pathogens [20]. The mechanisms exerted by biological control agents (BCAs) include direct (hyperparasitism, antibiosis) and indirect (competition, induction of host plant resistance) modes of action [19,21,22,23]. Direct suppression of target fungi by BCAs can be achieved by the production of antibiotics, volatile organic compounds (VOCs), or other bioactive secondary metabolites [21] that have an antagonistic effect on pathogens [19]. The inhibitory activity and biocontrol potential of some pure microbial non-volatile secondary metabolites against the wheat blast fungus MoT have been reported recently [24,25,26,27]. 

Bacterial VOCs are low-molecular-weight (<300 Da) [28,29], hydrophobic, low-toxicity, and naturally occurring substances [30]. They are diffusible in the environment and have been shown to be effective in biofumigation [31,32]. Recent discoveries of antifungal bacterial volatiles have triggered research interest in exploring the potential use of VOCs in the control of fungal pathogens. The efficiency of bacterial VOCs depends on the adaptability of bacteria to diverse environmental conditions, nutritional properties, and bacterial colonization patterns in specific hosts [21,33]. To date, relatively few bacterial biocontrol agents have been employed in agriculture due to their low field efficacy, safety concerns, or issues related to international market trading [34]. VOCs from *Pseudomonas* [30,35], *Bacillus* [36,37], *Burkholderia* [38,39], and *Serratia* [40] have been reported for their active antifungal activity. VOCs from *Trichoderma* fungi [41] are effective against wheat crown and root rot, and those from *Bacillus* spp. against wheat fusarium crown and root rot [42] and fusarium head blight [43]. 

The potential antagonistic activity of *B. subtilis* BTS-3, *B. velezensis* BTS-4, and *B. velezensis* BTLK6A against MoT was evaluated previously through in vitro and in vivo screening. These studies decoded the bacterial genes responsible for antagonism and induced systemic resistance (ISR) in host plants by whole-genome sequencing [44,45]. Additionally, the gene responsible for acetoin, a volatile organic compound, was confirmed in the tested *Bacillus* spp. However, it has been reported that *Bacillus* spp. can produce diversified antifungal volatile organic compounds (VOCs) to suppress phytopathogenic growth [36,37,42,43]. Thus, it is essential to identify all potential VOCs from our selected *Bacillus* spp. and their mechanisms of action to suppress the growth of MoT. Therefore, to further understand the mechanisms underlying the antagonistic action of these bacterial VOCs, this study investigates the role of *Bacillus* VOCs in the suppression of MoT without direct contact. More specifically, the objectives of the current study were to (i) investigate the effects of *Bacillus* VOCs on the mycelial growth and sporulation of MoT in vitro; (ii) demonstrate the suppressive effects of *Bacillus* volatiles against wheat leaf infection with MoT; (iii) identify and characterize *Bacillus* VOCs through GC–MS; and (iv) confirm the suppressive effects of selected pure VOCs against MoT.

## 2. Materials and Methods

### 2.1. Bacterial Strains and Culture Conditions

Three bacterial strains, *B. subtilis* BTS-3 (NCBI accession WOVJ00000000), *B. velezensis* BTS-4 (NCBI accession WOVK00000000), and *B. velezensis* BTLK6A (NCBI accession WOYD00000000), were used in this study [23]. BTS-3 and BTS-4 bacterial strains were isolated from the ‘Rangabinni’ rice seeds and BTLK6A from the ‘Kanchan’ wheat seeds of Bangladesh [23,44,46,47]. All *Bacillus* strains were stored as pure cultures in 20% glycerol at −20 °C. Bacterial strains were streaked into Petri dishes (90 mm) containing ca. 20 mL Luria broth agar (LBA: 10 g tryptone, 5 g yeast extract, 10 g NaCl, 15 g agar, 1000 mL H_2_O) and incubated for 24–48 h at 25 °C. Then, three single colonies were inoculated into a 50 mL Erlenmeyer flask containing 25 mL LB (10 g tryptone, 10 g NaCl, 5 g yeast extract, 1000 mL H_2_O) and incubated on a rotary shaker (100 rpm) for 24–48 h at 25 °C. After incubation, the bacterial cultures were transferred to 2 mL Eppendorf tubes and centrifuged for 10 min (13,000 rpm). The supernatant was discarded and the bacterial sediment washed (3 times) with sterilized distilled water (SDW). The bacterial densities were then adjusted (1 × 10^9^, 1 × 10^8^, and 1 × 10^7^ CFU/mL) for further use and stored in 20% glycerol at −80 °C for long-term preservation.

### 2.2. Fungal Strain and Culture Conditions

MoT fungal pathogen BTGP 6(f) was isolated from blast-infected wheat ear [48] and grown on V8 agar (V8A) following the protocol described by Surovy et al. [49]. The conidial suspension was prepared from 7-d-old MoT cultures by adding 0.01% sterile Tween 20 solution (10 mL) per plate. The suspension was filtered through a two-layer cheesecloth, and the conidial density was adjusted (1 × 10^5^ conidia/mL) using a hemocytometer (Fuchs-Rosenthal, 0.0625 mm^2^). 

### 2.3. Volatile Assays

#### 2.3.1. Bi-Partitioned Petri Dish Assay

Bi-partitioned Petri dishes (90 mm diameter) were used to assess the potential of *Bacillus* VOCs against MoT. LBA medium (10 mL) was poured into one side, and 10 mL of V8A was poured into the other side of the Petri dishes. The bacterial suspension (100 µL) was pipetted in LBA, spread with a glass spreader, and incubated for 24 h at 25 °C. Three different bacterial densities (1 × 10^9^, 1 × 10^8^, and 1 × 10^7^ CFU/mL) were used for this experiment. Twenty-four hours after bacterial incubation, a 2-mm 7-d-old MoT mycelial plug was placed on the side containing V8A. The Petri dishes were tightly closed with parafilm to avoid the evaporation of bacterial VOCs and incubated under the same conditions described earlier (see Section 2.2) for 5 d. The mycelial radial growth (mm) of MoT was recorded 5 d after incubation. Subsequently, 10 mL of sterilized 0.01% Tween 20 was added per plate and MoT conidia were dislodged from mycelia using a paint brush (da Vinci, Germany; size 3/0). The conidial suspension was filtered through a two-layer cheesecloth, and conidia were counted (conidia/plate) with a hemocytometer (Fuchs-Rosenthal, 0.0625 mm^2^). Six replications were maintained in each experiment, and three repetitive experiments were performed. 

#### 2.3.2. Upside-Down Petri Dish Assay

Bacterial strains at different densities (1 × 10^9^, 1 × 10^8^, and 1 × 10^7^ CFU/mL) were grown in Petri dishes containing LBA for 24 h at 25 °C. Twenty-four hours after bacterial incubation, 10 µL (1 × 10^5^ conidia/mL) of MoT conidial suspension was drop-inoculated in another Petri dish containing V8A. These two plates, one containing bacteria and one MoT, were placed face-to-face on top of each other, tightly sealed with parafilm to avoid the loss of VOCs, and incubated for 5 d at 25 °C. The mycelial radial growth of MoT (mm) and the total number of MoT conidia/plate were recorded as described in Section 2.3.1. Six replications were maintained in each experiment, and three repetitive experiments were performed.

### 2.4. Detached Leaf Assay

Wheat cultivar BR 18 was used for the detached leaf assay. The seeds were surface-sterilized with sodium hypochlorite (3% NaOCl) for 1 min and subsequently washed (3 times) with sterilized distilled water (SDW). Treated seeds were placed in Petri dishes containing moistened filter paper. After germination, they were then sown in plastic pots (7 × 7 × 8 cm; 10 seeds per pot) containing a mixture of sand, compost, and peat (1:2:1). Plants were grown in a greenhouse maintaining a 14/10 h light-dark cycle, 25 °C (±2) temperature, and 65–70% relative humidity. At growth stage 13 (GS 13, three leaves emerged), the second leaf was cut into small pieces (ca. 2 cm) and surface-sterilized with 3% NaOCl. The extra water from the surface-sterilized leaves was removed with a sterile paper towel. Leaf pieces were then placed on water agar (15 g agar, 1000 mL H_2_O) containing benzimidazole (30 mg/L). Ten leaf pieces were placed in each Petri dish. The MoT conidial suspension (1 × 10^5^ conidia/mL) was drop-inoculated (10 µL) on each leaf piece; for the control only, water (10 µL) was inoculated on each leaf instead of MoT conidial suspension. The bacterial suspension (100 µL, 1 × 10^9^ CFU/mL) was incubated in LBA for 24 h before the preparation of leaf pieces. After incubation, freshly grown (at 25 °C) bacterial culture plates were placed open and upside-down on the Petri dishes containing leaf pieces. Plates were sealed tightly with parafilm to avoid the loss of VOCs. Five days after incubation, the lesion growth and total number of conidia in each lesion were recorded. The lesion size (cm^2^) was determined by using the ImageJ software (version 1.53 m). A single leaf section was placed in a 2 mL Eppendorf tube containing 1 mL water, briefly vortexed, and MoT conidia per lesion were counted using a hemocytometer (Fuchs-Rosenthal, 0.0625 mm^2^). Thirty leaf pieces were used for each bacterial treatment, and three repetitive experiments were performed.

### 2.5. Identification and Quantification of Bacillus Volatiles

*Bacillus* VOC collection was performed as described previously by Sarenqimuge et al. [50]. As an internal standard, 200 ng of tetralin (1,2,3,4 tetrahydronaphthalene, Sigma-Aldrich, Munich, Germany) was added to each sample before GC–MS analysis. An aliquot of 30 µL sample was transferred to another GC vial with a glass insert and placed into the tray of the GC–MS autosampler. A 2 µL sample was injected in pulsed splitless mode for analysis. The oven temperature was retained at 40 °C for 3 min and gradually increased (8 °C/min) to a final temperature of 220 °C for 10 min. Helium was used as a carrier gas (flow rate was 1.5 mL/min). A homogenous series of n-Alkenes (C_7–20_) was used to determine retention indices. The MassHunter instrument (Agilent Technologies: GC 7890B, MS 5977B, Santa Clara, CA, USA) was used for data processing; MSD ChemStation software with the NIST17 and Willey11 mass spectral libraries was used to tentatively identify bacterial VOCs by their mass spectra and retention indices. The identities of the ten bioactive compounds tested in Section 2.6 were confirmed by GC–MS analysis of commercially available standards. The VOC quantification was performed by comparing the peak areas of individual compounds to the peak area of the internal standard (tetralin). From each treatment, five replicates were analyzed, and LB without bacteria was used as a control.

### 2.6. Bioassay with Pure Volatile Compounds

Pure VOCs (Appendix A) were tested against MoT at four different concentrations (5 M, 1 M, 500 mM, and 250 mM), with DMSO as a diluent. Five sterilized paper discs were glued (Tesa stick, tesa SE, Hugo-Kirchberg-Str.1, D-22848 Norderstedt) onto the Petri dish lid, and 20 µL of each pure compound was pipetted on each paper disc (total 100 µL per Petri dish). A 2 mm MoT mycelial block was placed in the center of a V8A plate and the two plates were sealed tightly to avoid the loss of VOCs. Five days after incubation, the mycelial radial growth of MoT and the total number of conidia per plate were recorded as described in Section 2.3.1.

### 2.7. Statistical Analysis

All data were analyzed by using linear models (LMs) in the R software (version 4.0.5, accessed 31 March 2021) integrated into R studio (version 1.2.5001, accessed 31 March 2021). The functions ‘test dispersion’ and ‘StimulatedResiduals’ of the ‘DHARMa’ package were used to test the dispersion and residuals of the models. These functions of the ‘DHARMa’ package use a simulation-based method to create readily interpretable, scaled residuals for fitted linear models. Analysis of variance (ANOVA) was calculated for normally distributed data, followed by Tukey multiple comparisons (*p* < 0.05), by using the ‘emmeans’ package. For non-normally distributed data sets, the Kruskal–Wallis test was performed by using the ‘kruskal.test’ function, followed by Dunn multiple comparison analyses by using the ‘FSA’ and ‘rcompanion’ packages (*p* < 0.05). The ‘ggplot2’ package was used to visualize bar graphs, and the ‘ggVennDiagram’ function was used to plot the number of VOCs produced in different *Bacillus* treatments in a Venn diagram. The ‘ComplexHeatmap’ function was used to visualize the *Bacillus* VOC profiles, the effects of pure VOCs on MoT mycelial growth, and MoT sporulation in a heatmap. MetaboAnalyst 5.0 [51] was used for volcano plot analysis.

## 3. Results

### 3.1. Bioassay with Volatiles

#### 3.1.1. Effects of *Bacillus* VOCs on MoT Growth

Three different bacterial densities (1 × 10^7^, 1 × 10^8^, and 1 × 10^9^ CFU/mL) were used in a bi-partitioned Petri dish assay to evaluate the effects of *Bacillus* VOCs on MoT mycelial growth. Different bacterial VOCs significantly inhibited MoT mycelial growth in a density-dependent manner (F = 161.44, *p* < 0.001) (Figure 1A). Compared to the control (43 mm), the highest reduction (72.5%) in mycelial growth was recorded in BTS-4 (1 × 10^9^ CFU/mL, 11.8 mm diameter), and the lowest (3%) in BTLK6A (1 × 10^7^ CFU/mL and 41.5 mm diameter). Inhibition of MoT mycelial growth was consistently higher at the bacterial density of 1 × 10^9^ CFU/mL for all four *Bacillus* treatments (Figure 1A,B).

In addition to the assessment of MoT mycelial growth, the sporulation rate was evaluated after 5 d of bacterial treatment. Treatments with *Bacillus* VOCs significantly suppressed MoT sporulation (F = 23.092, *p* ≤ 0.001). All three densities (1 × 10^7^, 1 × 10^8^, and 1 × 10^9^ CFU/mL) of BTS-3, BTS-4, and the *Bacillus* consortium treatments produced non-spore-forming white mycelia (100% reduction in conidia compared to control). In the case of BTLK6A, no sporulation was observed at 1 × 10^9^ CFU/mL. However, at 1 × 10^7^ and 1 × 10^8^ CFU/mL densities, black/grey-colored MoT sporulating colonies were observed as in the untreated control. However, the number of conidia was comparatively less in 1 × 10^7^ (2.37 × 105 conidia/plate, 73% reduction compared to control) and 1 × 10^8^ CFU/mL (6.60 × 10^4^ conidia/plate, 92% reduction compared to control) of BTLK6A treatment compared to the control (9.10 × 10^5^ conidia/plate) (Figure 1C). 

#### 3.1.2. Effects of *Bacillus* VOCs against Germination of MoT Conidia 

An upside-down Petri dish assay was performed to evaluate the effect of *Bacillus* VOCs on MoT conidia germination. All MoT conidia germinated, and mycelial growth ensued after exposure to *Bacillus* VOCs. However, MoT mycelial growth was very slow in the *Bacillus* consortium treatment (1 × 10^9^ CFU/mL). Additionally, less intense, flat mycelial growth was observed with all BTS-4 treatments (Figure 2A). Similar to the bi-partitioned Petri dish assay, the bacterial VOCs also significantly inhibited MoT mycelial growth (developed from MoT conidia) (F = 372.63, *p* ≤ 0.001). 

Mycelial growth reduction was higher in the upside-down Petri dish assay than the bi-partitioned Petri dish assay. The highest inhibition of mycelial growth was recorded with the treatment of the *Bacillus* consortium (1 × 10^9^ CFU/mL), with radial growth of 13.2 mm, followed by BTS-4 (1 × 10^9^ CFU/mL, 16.8 mm). The lowest reduction was documented for BTLK6A (1 × 10^7^ CFU/mL, 42.9 mm) (Figure 2B). The highest sporulation was recorded for the control (9.73 × 10^5^ conidia/plate). However, the complete suppression of sporulation of MoT was recorded for all bacterial treatments except for BTLK6A (1.23 × 10^5^ conidia/plate at 1 × 10^7^ CFU/mL, 87% reduction in sporulation compared to control) (Figure 2C).

### 3.2. Effects of Bacillus VOCs in Detached Leaf Assay

To investigate the capacity of *Bacillus* VOCs to reduce leaf infection with MoT, a detached leaf assay was performed using four different *Bacillus* VOC treatments (Figure 3A). In our experiment, we found that *Bacillus* VOCs significantly reduced the development of blast disease symptoms in detached leaves and suppressed MoT sporulation under laboratory conditions, but with varying effects. *Bacillus* VOCs significantly reduced the lesion size (F = 37.14, *p* ≤ 0.001) and MoT conidia production (F = 28.49, *p* ≤ 0.001) (Figure 4A). The largest lesion (0.48 cm^2^) was recorded in the untreated control, followed by BTLK6A, where BTLK6A VOCs reduced the lesion size by 43.75% (0.27 cm^2^) compared to the control. The smallest lesion was observed in BTS-4, with a >85% reduction in lesion size (0.07 cm^2^) compared to the control, followed by the *Bacillus* consortium (81.25% reduction, 0.09 cm^2^) and BTS-3 (72.9% reduction, 0.13 cm^2^). Therefore, there were no significant differences between the BTS-3, BTS-4, and *Bacillus* consortium treatments (Figure 3B). 

MoT sporulation in VOC-treated leaf lesions was lower compared to the control. A single lesion on a control leaf segment yielded 5.6 × 10^4^ conidia/lesion, significantly different from all other bacterial treatments. The BTS-4-treated leaf segments had the lowest number of conidia (1.9 × 10^3^ conidia/lesion), followed by the *Bacillus* consortium (3.5 × 10^3^ conidia/lesion). However, the numbers of conidia produced in BTS-4 and consortium-treated leaf lesions were not significantly different. There was no MoT sporulation in the water-treated control as there was no MoT infection present (Figure 3C).

### 3.3. Identification and Quantification of Bacillus Volatile Organic Compounds (VOCs)

The VOCs produced from the four different *Bacillus* treatments (BTS-3, BTS-4, BTLK6A, and consortium (a mixture of all three *Bacillus* strains)) were identified and quantified using GC–MS. Thirty-nine VOCs were identified in total, of which 11 were produced by all four bacterial treatments (Figure 4A, Appendix A). 

The greatest diversity of VOCs were released by BTS-4 (34), followed by the *Bacillus* consortium (22), and lastly by BTLK6A (12). Among the 39 VOCs, 12 unique VOCs were produced by BTS-4, two by the *Bacillus* consortium, and only one by BTS-3. A total of nine different classes of volatiles were identified: alkanes 7.70%, alcohols 23.07%, ketones 20.51%, fatty acids 12.82%, aldehydes 15.38%, aromatic 2.56%, N-containing 10.25%, S-containing 5.12%, and alkene compounds 2.56% (Figure 4B). Alcohol, fatty acid, ketone, S-containing, and aldehyde compounds were identified in all four *Bacillus* treatments. The highest number of diversified VOC classes was detected in BTS-4 (9), followed by BTS-3 (8), the consortium (7), and BTLK6A (6). The number of alcoholic VOCs was higher for BTS-4, followed by the *Bacillus* consortium. Fatty acid VOCs were also higher in BTS-4, followed by the BTS-3 and *Bacillus* consortium treatments (Figure 4B). The concentrations of bacterial VOCs produced by different treatments differed significantly. The heatmap analysis represents the VOC clustering and the relationships between different bacterial treatments (Figure 4C). 

In vitro and in vivo experimental data indicated that VOCs from the BTS-4 and *Bacillus* consortium treatments had considerable potential to control MoT. Therefore, we investigated the relationships between the VOCs produced from the BTS-4 and consortium treatments to determine the effectiveness of BTS-4 and *Bacillus* consortium volatiles against MoT. Figure 5 displays the fold change (*p* ≤ 0.05) in VOC production from BTS-4 compared to the *Bacillus* consortium. In BTS-4, 21 VOCs were upregulated, 4 were down-regulated, and 9 were not significantly different from the *Bacillus* consortium treatment (Figure 5). 

### 3.4. Effect of Pure VOCs on Mycelial Growth and Sporulation of MoT

From 39 identified VOCs, ten bioactive VOCs (2-methyl propionic acid, 3-methyl-1-butanol, 3-methyl butanoic acid, 2,3-butanediol, and 2-methyl butanoic acid, phenyl ethyl alcohol, hexanoic acid, 2-methyl-1-butanol, 2,5-dimethyl pyrazine, and acetoin) were tested in vitro against MoT based on the previous literature [50]. DMSO and water were used as positive controls. Of the selected compounds, four VOCs (2-methyl butanoic acid, 2-methyl propanoic acid, 2,5- dimethyl pyrazine, and 3-methyl butanoic acid) were produced by all four bacterial treatments; three (2,3-butanediol, 3-methyl-1-butanol, and phenyl ethyl alcohol) were produced by BTS-4 and the *Bacillus* consortium; two VOCs (2-methyl-1-butanol and acetoin) only in BTS-4; and one (hexanoic acid) in the BTS-3 and BTS-4 treatments.

The efficacy of single pure volatile compounds against MoT mycelial growth was assessed in an in vitro bioassay. Pure VOCs were used at four different concentrations (5 M, 1 M, 500 mM, and 250 mM). At 5 M, all compounds except acetoin and 2,3-butanediol inhibited the mycelial growth of MoT (Figure 6A). Among the pure compounds, hexanoic acid suppressed MoT growth up to a 500 mM concentration (Appendix A).

In parallel with the reduction in MoT mycelial growth, the selected pure VOCs significantly reduced sporulation from MoT mycelia in vitro (Appendix A). No sporulation was observed in any of the four treatments with phenylethyl alcohol (PEA). Similarly, no sporulation was recorded for hexanoic acid or 2-methylbutanoic acid up to a 500 mM concentration (Figure 6B). Figure 6A,B contain heatmaps showing the relative effects of potential VOCs on the reduction in the mycelial growth and sporulation of MoT. The lowest VOC concentration inhibitory to MoT sporulation was recorded for PEA (250 mM), followed by 2-methylbutanoic acid and hexanoic acid at 500 mM (Figure 6B).

## 4. Discussion

In this study, we used four bacterial treatments (BTS-3, BTS-4, BTLK6A, and a consortium of *Bacillus* spp.) to assess the effects of *Bacillus* spp. VOCs to control an emerging fungal pathogen, the *M. oryzae Triticum* (MoT) pathotype. All *Bacillus* treatments produced diverse VOCs and exhibited strong antagonism against MoT, by suppressing mycelial growth and sporulation in vitro. It is well documented that *Bacillus* VOCs exert antifungal activity against various phytopathogens [52,53]. Additionally, some studies have reported that the volatiles from *B. megaterium* [54], endophytic *Chryseobacterium* [37], and *Pseudomonas* sp. [55] suppress the mycelial growth of the rice blast pathogen (*M. oryzae Oryzae* pathotype). However, so far, no information has been made available about the suppression of MoT mediated by *Bacillus* volatiles. To the best of our knowledge, this is the first report on *Bacillus* VOCs significantly inhibiting the mycelial growth of this important pathogen.

It has been previously reported that bacterial consortia are more effective at controlling certain fungal pathogens than single bacterial strains [56,57]. In this study, *B. velezensis* BTS-4 and the *Bacillus* consortium performed better than the other *Bacillus* treatments in suppressing MoT mycelial growth and conidial germination in vitro. Additionally, the densitiy of *Bacillus* spp. had a significant positive correlation with MoT inhibition. At 1 × 10^9^ CFU/mL, the MoT inhibition rate was higher than at 1 × 10^7^ and 1 × 10^8^ CFU/mL. The higher densitiy of *Bacillus* spp. led to more *Bacillus* colony growth, higher VOC production, and a significant reduction in the growth and sporulation of MoT. *B. velezensis* can inhibit the growth of *Colletotrichum gloeosporioides* at a density of 1 × 10^7^ CFU/mL [52] and *S. sclerotiorum* at a density of 1 × 10^8^ CFU/mL [58]. Furthermore, *B. subtilis* has been documented to control *Alternaria solani* at a density of 1 × 10^8^ CFU/mL [59], and *B. amyloliquefaciens* VOCs can suppress *Fusarium oxysporum* f. sp. *cubense* in vitro also at a density of 1 × 10^8^ CFU/mL [60].

These findings suggest that the VOCs from *Bacillus* spp. may lead to the functional degradation of MoT mycelia and thus suppress sporulation from MoT mycelia. Deformed hyphae with vacuolation, excessive branching, the degeneration of hyphal cells, or combinations of excessive branching with vacuolation were recorded (Appendix A). Likewise, the VOCs from *B. velezensis* and *B. atrophaeus* also cause vacuolation and cavities in the mycelial cytoplasm of *B. cinerea* [58], and VOCs of *B. subtilis* may cause expanded, uneven, flaccid hyphae and the suppression of *A. solani* sporulation [59].

Furthermore, *Bacillus* VOCs significantly reduced the leaf blast lesion size and further MoT sporulation from the lesions in an in vivo detached leaf assay. Conidia are the main dispersal units of MoT epidemiology. Reduced or no conidia formation will result in reduced wheat infection by MoT. Therefore, understanding the MoT sporulation reduction mechanisms of *Bacillus* VOCs is the first step in controlling MoT epidemics. Earlier reports reveal that the volatiles of *Bacillus* spp. May reduce the lesion size and sporulation of *A. solani* in potato leaves [59], as well as the sporulation of *Sclerotinia sclerotiorum* on tomato, tobacco, and soybean leaves in vivo [61]. However, our results confirm that exposure to *Bacillus* VOCs does not entirely prevent MoT infection or sporulation but rather slows down the development of blast symptoms compared to the untreated control. *Bacillus* VOCs also significantly reduced blast lesion development in the detached spike assay (data not shown).

GC–MS was used to determine the active VOCs from different *Bacillus* treatments. Thirty-nine VOCs were identified in total, of which 11 VOCs were produced in all four *Bacillus* treatments. The emitted bacterial VOCs were alcohols, alkenes, alkynes, ketones, aldehydes, fatty acids, aromatic, N-containing, and S-containing compounds. The VOCs identified in our analysis have demonstrated broader antifungal activity against phytopathogens. The mixture of alcoholic volatiles 2-methyl-1-butanol and 3-methyl-1-butanol was very effective in suppressing the growth of *Phyllosticta citricarpa* [62] and *Aspergillus flavus* [63], and 2-ethyl-1-hexanol strongly inhibited the growth of *Colletotricum acutatum* [64] and *B. cinerea* [65].

The compound 6-methyl-2-heptanone disrupts mycelial integrity, collapses conidial vesicles, and downregulates the conidial germination gene of pathogenic fungus [66]. The S-containing volatile compound benzothiazole inhibits cystospore germination and the mycelial growth of *Phythophthora parasitica* var. *nicotianae* [67]. Hexanal, an aldehyde group volatile, induces systemic resistance in mango plants by inducing defense-related enzymes (phenylalanine ammonia lyase (PAL), peroxidase (PO), polyphenol oxidase (PPO), superoxide dismutase (SOD), and catalase (CAT)), thus significantly reducing *Lasiodiplodia theobromae* infection [68].

In our study, ten VOCs were selected to test their potential in inhibiting MoT based on their reported bioactivity in the literature. All selected VOCs except acetoin and 2,3-butanediol significantly reduced MoT mycelial growth and sporulation. Acetoin and 2,3-butanediol play a role in inducing systemic resistance in plants [69,70] and do not seem to be directly involved in the suppression of MoT mycelial growth and sporulation. Meanwhile, 2,5-dimethyl pyrazine stopped MoT sporulation at a 1 M concentration and has also been cited to control *Sclerotinia* sp., *Pythium* sp., *Rhizoctonia* sp. [71], and *Anthracnose* sp. [72]. The activity of 2-methyl propanoic acid against MoT was not promising; although it effectively controls rubber white root rot disease, it negatively affects seedling growth [73].

Hexanoic acid, phenylethyl alcohol, and 2-methyl butanoic acid potentially inhibited the mycelial growth and sporulation of MoT. Phenylethyl alcohol slows phytopathogenic growth by inhibiting the synthesis of RNA, DNA, and protein and upregulates genes related to the phagosome, peroxisome, proteasome, and autophagy [74]. Considering this information, it can be deduced that the phenylethyl alcohol first causes MoT mycelial alternations and later induces autophagy, triggering programmed cell death. Fatty acids have also been reported to exert inhibitory activity against some fungal pathogens, but saturated fatty acids have robust antifungal activity compared to other fatty acids [75]. This study found that hexanoic acid (a saturated fatty acid) inhibited mycelial growth and sporulation up to 500 mM. It has been documented that the minimum inhibitory concentrations (MICs) of hexanoic acid against *Micosporum gypseum* range from 0.02 to 75 µg/mL [76]. At a concentration of 10 mM, *Candida albicans* growth is inhibited by hexanoic acid through changes in intracellular hydrostatic pressure and subsequent disruption of the cell plasma membrane [77]. Additionally, hexanoic acid enhances plant jasmonic acid (JA) signaling and induces callose deposition during fungal infection [78].

Organic fertilizers promoted the growth of *B. amyloliquefaciens*, induced the release of 2-nonanone and nonanal, and suppressed *R. solanacearum* [79]. The encapsulation of *Bacillus* VOCs might be an effective way to use bacterial VOCs under field conditions; thus, it facilitates the slow and steady release of volatiles. Effective control of MoT using bacterial VOCs requires more detailed studies considering field environmental conditions and compatability with other control strategies. Therefore, our study suggests that *Bacillus* VOCs are potential biologicals to suppress MoT, with fundamental and practical implications for wheat production through reducing the severity of wheat blast. As *Bacillus* spp. are rich in the production of both volatile and non-volatile antimicrobial compounds [47], further studies are warranted to identify non-volatile antimicrobial secondary metabolites from the investigated *Bacillus* spp. that might work together to effectively control wheat blast. Field evaluation of wheat blast suppression by these *Bacillus* and their metabolites is required before recommending them for practical application in the biorational management of wheat blast.

## 5. Conclusions

*Bacillus* produces diverse antifungal volatile organic compounds (VOCs) that are able to suppress the growth and sporulation of MoT conidia in vitro and in vivo. Wheat blast is mainly caused by infections initiated by MoT conidia, and the suppression of MoT sporulation may have practical relevance and fundamental implications in reducing wheat blast severity.

## Figures and Tables

**Figure 1 microorganisms-11-01291-f001:**
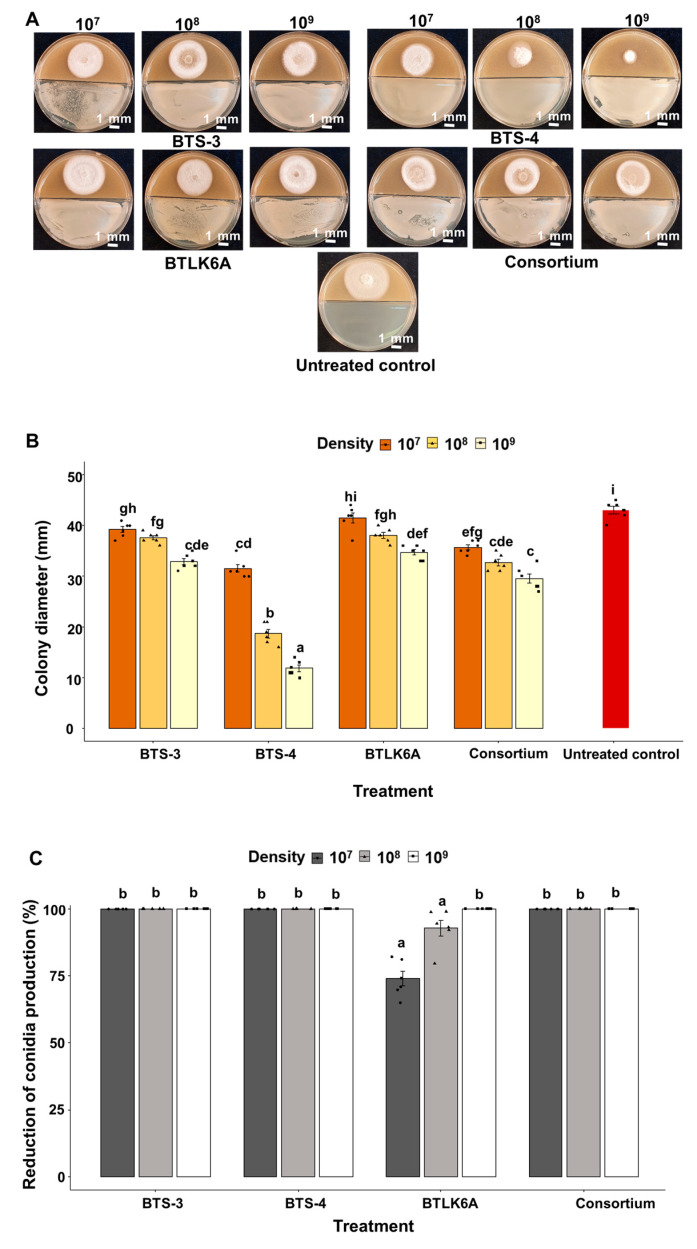
Effect of *Bacillus* spp. VOCs on MoT in bi-partitioned Petri dish assay in vitro. (**A**) Photographs showing the effects of *Bacillus* VOCs on mycelial growth of MoT; (**B**) effects of *Bacillus* VOCs on mycelial growth of MoT in vitro; (**C**) effects of *Bacillus* VOCs on reduction in MoT sporulation compared to control in vitro. Data were recorded after 5 d of MoT incubation at 25 °C. BTS-3: *B. subtilis* BTS-3; BTS-4: *B. velezensis* BTS-4; BTLK6A: *B. velezensis* BTLK6A; consortium: a mixture of BTS-3, BTS-4, and BTLK6A; control: without any bacterial treatment (Tukey test; *n* = 6; *p* ≤ 0.05). Black points in (**B**,**C**) represent data points for each replicate.

**Figure 2 microorganisms-11-01291-f002:**
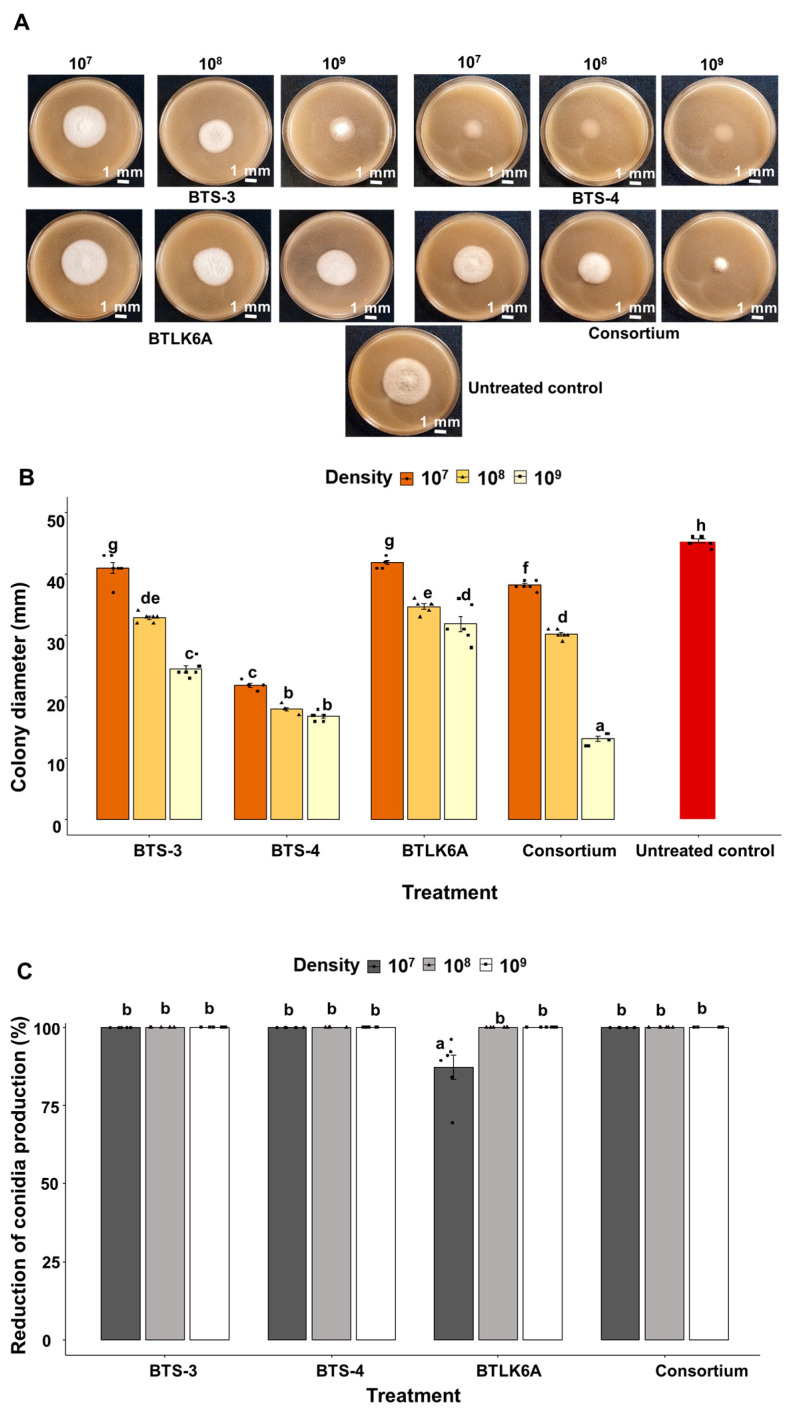
Effects of *Bacillus* spp. VOCs on MoT conidial germination in upside-down Petri dish assay in vitro. (**A**) Photographs showing the effects of *Bacillus* VOCs on MoT conidial germination and mycelial growth; (**B**) effects of bacterial VOCs on mycelial growth of MoT in vitro; (**C**) effects of bacterial VOCs on reduction in new MoT sporulation compared to control in vitro. Data were recorded after 5 d of MoT incubation at 25 °C. BTS-3: *B. subtilis* BTS-3; BTS-4: *B. velezensis* BTS-4; BTLK6A: *B. velezensis* BTLK6A; consortium: a mixture of BTS-3, BTS-4, and BTLK6A; control: without any bacterial treatment (Tukey test; *n* = 6; *p* ≤ 0.05). Black points in (**B**,**C**) represent data points for each replicate.

**Figure 3 microorganisms-11-01291-f003:**
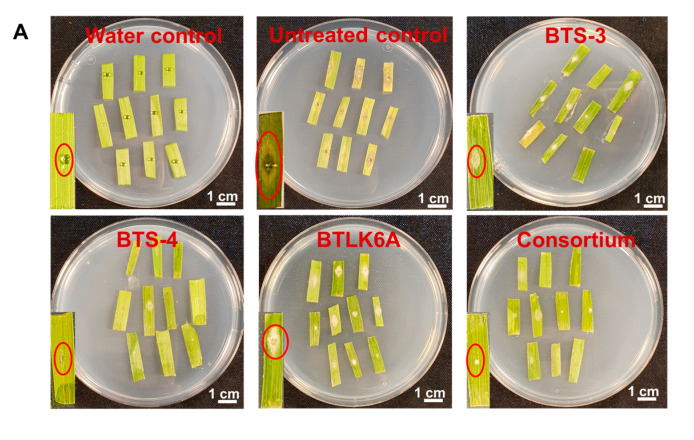
Effects of *Bacillus* VOCs on lesion development and MoT sporulation in a detached wheat leaf assay (cv. BR-18). (**A**) VOCs from *Bacillus* spp. significantly reduced leaf blast lesion size in vivo. (**B**) *Bacillus* VOCs significantly reduced leaf lesion area (cm^2^) caused by MoT in vivo (Kruskal–Wallis test; *n* = 30; *p* ≤ 0.05); (**C**) reduction in MoT sporulation by VOCs from different *Bacillus* spp. in vivo (Tukey test; *n* = 30; *p* ≤ 0.05). Data were recorded after 5 d of MoT incubation at 25 °C. BTS-3: *B. subtilis* BTS-3; BTS-4: *B. velezensis* BTS-4; BTLK6A: *B. velezensis* BTLK6A; consortium: a mixture of BTS-3, BTS-4, and BTLK6A; untreated control: only MoT inoculated. Black points in (**B**,**C**) represent data points for each replicate.

**Figure 4 microorganisms-11-01291-f004:**
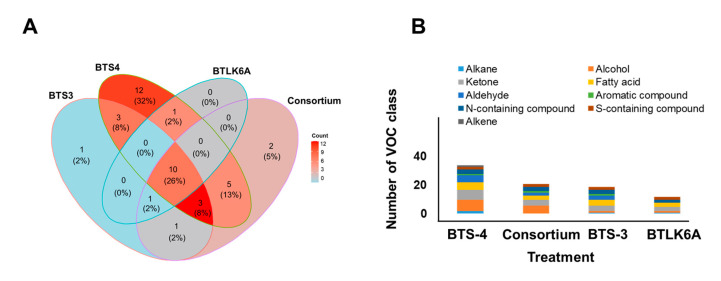
Identification and quantification of VOCs from different *Bacillus* treatments by GC–MS analysis. (**A**) Venn diagram representing the number of VOCs produced from different *Bacillus* spp.; (**B**) number of different VOC classes produced from different *Bacillus* treatments; (**C**) heatmap based on Euclidean distance showing the VOCs produced from each *Bacillus* treatment. Each line in the color heatmap indicates a single compound; red to green color code indicates low to high relative concentrations (based on row Z-scores) of the compounds; blue color indicates compounds detected only in a single treatment; grey color indicates undetected volatile. BTS-3: *B. subtilis* BTS-3; BTS-4: *B. velezensis* BTS-4; BTLK6A: *B. velezensis* BTLK6A; const.: consortium (a mixture of *B. subtilis* BTS-3, *B. velezensis* BTS-4, and *B. velezensis* BTLK6A) (*n* = 5; *p* ≤ 0.05). The VOCs were quantified from 40 mL LB inoculated with 100 µL (1 × 10^9^ CFU/mL) bacteria and incubated for 4 d at 25 °C.

**Figure 5 microorganisms-11-01291-f005:**
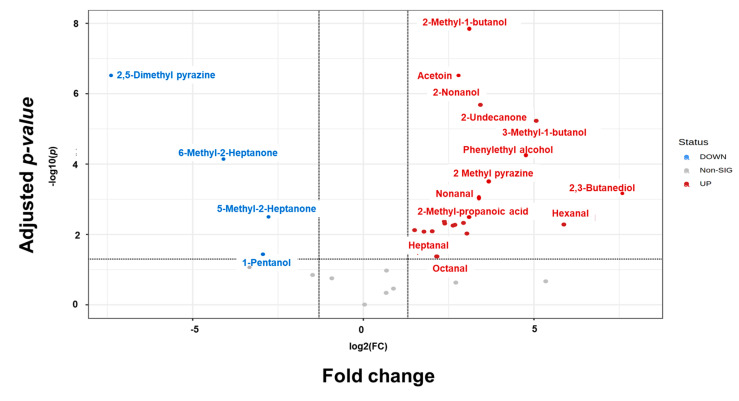
Fold change (FC) in VOCs produced by *B. velezensis* BTS-4 in relation to the *Bacillus* consortium (a mixture of *B. subtilis* BTS-3, *B. velezensis* BTS-4, and *B. velezensis* BTLK6A) treatment through volcano plot analysis. The log2 fold change threshold was 2.0. The false discovery rate (FDR) was maintained with *p* threshold at 0.05. The VOCs were quantified from 40 mL LB inoculated with 100 µL (1 × 10^9^ CFU/mL) bacteria and incubated for 4 d at 25 °C. Non-SIG: non-significant.

**Figure 6 microorganisms-11-01291-f006:**
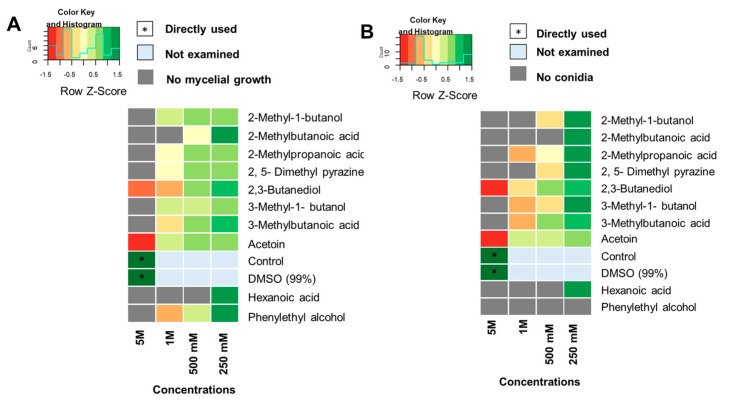
Effect of selected pure antifungal VOCs against MoT mycelial growth and sporulation in vitro represented by heatmaps based on Euclidean distance: (**A**) effect of pure antifungal VOCs on MoT mycelial growth; (**B**) effect of pure antifungal VOCs on MoT sporulation. Each row in the color heatmap indicates a single compound. Red to green color code indicates low to high relative growth or sporulation of MoT (based on row Z-score). * indicates directly used single concentration (diluent) (*n* = 6, *p* ≤ 0.05).

## Data Availability

All data supporting the results are included in the manuscript.

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
