# Peer review of "Suppressive Effects of Volatile Compounds from Bacillus spp. on Magnaporthe oryzae Triticum (MoT) Pathotype, Causal Agent of Wheat Blast"

_microorganisms, 2023, doi:10.3390/microorganisms11051291_

Round 1

Reviewer 1 Report

This scientific study offers new information regarding VOCs from Bacillus species as potent inhibitors of Magnaporthe oryzae Triticum, the agent responsible for the wheat blast. The paper was written in simple terms, and it thoroughly analyzed the available scientific papers. There are a few small criticisms, though:

1.     L111 In the phrase "The supernatant was discarded and washed ...", the word sediment should probably be added.

2.     Untreated control is not present in Figures 1 and 3.

3.     The placement of Figures 1, 3 and 4 is unsuccessful. Figures must be placed entirely on the page.

4.     Ls 299 – 300 What is water control? Its preparation should be included in the Materials and Methods section.

The article is written in an accessible language, but you need to check the punctuation

Author Response

Thank you for your valuable comments.

1.L111 In the phrase "The supernatant was discarded and washed ...", the word sediment should probably be added.

Response: Changed accordingly.

2. Untreated control is not present in Figures 1 and 3.

Response: Changed accordingly

3. The placement of Figures 1, 3 and 4 is unsuccessful. Figures must be placed entirely on the page.

Response: Changed accordingly

4. Ls 299 – 300 What is water control? Its preparation should be included in the Materials and Methods section.

Response: Water control = Only water is inoculated on leaf pieces instead of MoT conidial suspension. We added the description of water control in Material and Methods section accordingly.

Reviewer 2 Report

According to the manuscript, this important study investigated the Bacillus VOCs for inhibiting Magnaporthe oryzae Triticum (MoT) that causes the wheat blast disease. This manuscript is recommended for publication in the Microorganisms Journal after minor revision. Please find the reviewer’s comments below.

1. As Fig. 1, why does the Bacillus consortium show little effect on the inhibition of MoT compared with the results of the upside-down Petri dish assay (Fig. 2)?

2. Why were the VOCs produced from the Bacillus consortium different from the single bacterial strains (BTS-3, BTS-4, and BTLK6A)?

3. Please add the conclusion section to summarize the findings of this study.

Author Response

Thank you for your valuable comments.

1. As Fig. 1, why does the Bacillus consortium show little effect on the inhibition of MoT compared with the results of the upside-down Petri dish assay (Fig. 2)?

Response: In Fig. 1, we used mycelial plugs of MoT and Bacillus which were placed on the opposite sides of Petri dishes. However, in Fig. 2, we used MoT conidia instead of mycelial plugs. In case of Fig. 2, Bacillus volatiles may have inhibited/or slowed down the growth of MoT conidia more efficiently, as the target fungus was more directly exposed to the VOCs. In addition, we assume that the upsite down assay may have resulted in higher VOC concentrations to which MoT conidia were exposed. 

2. Why were the VOCs produced from the Bacillus consortium different from the single bacterial strains (BTS-3, BTS-4, and BTLK6A)?

Response: The Bacillus consortium is the mixture of all three bacterial strains. When Bacillus strains are grown in a consortium, there might be bacterial competition for space and nutrient acquisition. We assume that this may alter the composition of volatiles produced by the consortium as compared to single bacterial strains.

3. Please add the conclusion section to summarize the findings of this study.

Response: Added accordingly.

Reviewer 3 Report

Dear Authors,

The subject of the study is interesting and topical, with high scientific and practical importance.

The introduction is presented correctly, in accordance with the subject. Numerous scientific articles, in concordance to the topic of the study, were consulted.

Methodology of the study was clearly presented, and appropriate to the proposed objectives.

The obtained results are important and have been analyzed and interpreted correctly, in accordance with the current methodology.

The discussions are appropriate, in the context of the results, and was conducted compared to other studies in the field.

The scientific literature, to which the reporting was made, is recent and representative in the field.

Some minor (insignificant) suggestions and corrections were made in the article.

Minor aspects, regarding the writing of a unit of measure (e.g. mm2) and spaces between some words and the bibliographic sources

Author Response

Thank you for your review and comments. We revised our manuscript accordingly.
